# Real-Time Quantitative PCR Analysis of the Expression Pattern of the Hypoglycemic *Polypeptide-P* Gene in *Momordica charantia*

**DOI:** 10.3390/genes10121044

**Published:** 2019-12-16

**Authors:** Yi-Shuai Wang, Xiang-Qing Zeng, Xu-Zhong Yang, Wei Liu, Peng-Fei Li, Fu-Jun Wang, Jian Zhao

**Affiliations:** 1State Key Laboratory of Bioreactor Engineering, East China University of Science and Technology, Shanghai 200237, China; wys19900618@163.com (Y.-S.W.); pengfeili@ecust.edu.cn (P.-F.L.); 2Institute of Chinese Materia Medica, Shanghai University of Traditional Chinese Medicine, 1200 Cailun Road, Shanghai 201203, China; zxqabc@126.com (X.-Q.Z.); wfj@shutcm.edu.cn (F.-J.W.); 3Zhejiang Reachall Pharmaceutical Co. Ltd., 209 West Hulian Road, Dongyang City 322100, China; xuzhong.yang@hotmail.com

**Keywords:** *Momordica charantia*, hypoglycemic *polypeptide-P*, qPCR, Western blotting, expression pattern

## Abstract

This study was designed to establish a real-time quantitative polymerase chain reaction (qPCR) method to rapidly and reliably analyze the hypoglycemic *polypeptide-P* gene expression pattern in *Momordica charantia* (MC) and to examine its expression changes in different MC accessions, harvesting seasons, and tissue types. The qPCR results were further verified by using Western blotting (WB). A total of 10 MCs with different accessions were collected and tested in this study. Among the tested accessions, RU5H showed the highest expression level of the *polypeptide-P* gene. The expression level of the *polypeptide-P* gene was not only season-related (with the highest in early July) but also tissue-related (with the highest in the seed tissue). In addition, the expression characteristic of the *polypeptide-P* gene was maturity-related, with the highest expression level in the tender MC. The WB results show that the transcription level of this gene shows an almost similar trend to the corresponding protein expression level. In conclusion, the established qPCR method can rapidly and effectively detect the expression levels of the *polypeptide-P* gene in MCs with different accessions; furthermore, various factors, including the accessions, harvesting seasons, and tissue types can affect the expression level.

## 1. Introduction

Diabetes mellitus (DM), generally referred to as diabetes, is an endocrine disorder with the symptom of persistent high blood glucose levels caused by the decrease or defect of insulin [1,2]. At present, DM has become the third leading cause of death following cardiovascular diseases and cancers [1,2,3]. 

In general, patients with DM need to keep taking drugs throughout their life, and the DM treatment requires a higher security standard. Thus, it is of great importance to develop novel anti-diabetic drugs with few side effects [4,5]. 

*Momordica charantia* (MC) has been traditionally used for diabetic treatment in Southeast Asia and China due to its antihyperglycemic and antioxidant activities [3,6,7,8,9], and the remarkable hypoglycemic function of MC has also been widely acknowledged [10,11]. Accumulating evidence has demonstrated that MC polypeptide is the main active ingredient and that it plays a critical role in the hypoglycemic function [12,13,14,15,16]. Many hypoglycemic polypeptides have been isolated from certain Indian MC varieties, and a water-soluble polypeptide component was named polypeptide-P [14]. The expression levels of various bio-active ingredients among MCs with different accessions, including some polypeptides, vary dramatically, and this can result in a significant difference in antihyperglycemic activity [5,17].

A previous study showed that the content of polypeptide-P exhibits distinct differences in various MC varieties [18]. Therefore, selecting suitable MC accessions with high hypoglycemic activity is pivotal for patients with diabetes to relieve symptoms and to reduce medication costs. The traditional method of selecting MC accession with high hypoglycemic activity is to purify polypeptide-P from different MC varieties and then compare the relative content [19]. However, because of the low content of polypeptide-P in plants, this screening process is usually time-consuming and laborious. 

A real-time quantitative polymerase chain reaction (qPCR) is a potential method for the rapid detection of a target gene expression level and has been widely used in species identification [20,21,22,23]. In 2011, the nucleotide sequence of the *polypeptide-P* gene from MC was successfully cloned [24]. To understand the expression pattern of the hypoglycemic *polypeptide-P* gene, and to identify the key factors affecting the expressed levels in different MC varieties, we established a qPCR procedure in this study to analyze the transcription level of the MC *polypeptide-P* gene and further analyzed the corresponding protein expression level by using Western blotting (WB). Our results show that the expression level of the *polypeptide-P* gene in MC is affected by many factors, such as the accessions, harvesting seasons, and tissue types.

## 2. Materials and Methods 

### 2.1. Plant Materials

A total of 10 MCs with different accessions were collected in this study (Table 1). They were grown in natural environments. Mature fruits were respectively collected in the summer (June and July) and autumn (September and October) of 2015 and were immediately frozen at −80 °C.

### 2.2. Total RNA Extraction and cDNA Synthesis

The total RNA was extracted from each frozen MC sample (approximately 100 mg) based on the use of hexadecyltrimethyl ammonium bromide (CTAB), polyvinylpyrrolidone (PVP), and β-mercaptoethanol [25]. To remove any DNA contamination, the extracted RNA was treated with RNase-free DNase (TaKaRa, Dalian, China). The concentrations of each RNA sample were measured using a U-2900 double beam spectrophotometer (Hitachi, Tokyo, Japan). Only the RNA samples with an OD_260 nm_/OD_280 nm_ ratio between 1.9 and 2.1, as well as those with an OD_260 nm_ /OD_230 nm_ ratio greater than 2.0, were used for the analysis. The integrity of RNA samples was also assessed by 1% agarose gel electrophoresis. Subsequently, 1 μg of total RNA was used to synthesize cDNA in a total volume of 20 μL by using the reverse transcriptase (TaKaRa, Dalian, China), and the 10-fold diluted cDNA was used as a template to quantify the target gene expression level by using qPCR.

### 2.3. Protein Isolation

A protocol described previously was used for protein isolation from MC [18]. Briefly, an equal mass of MC sample (approximately 120 mg) was ground in liquid nitrogen to yield a fine powder. The powder was quickly transferred to a tube containing phosphate-buffered saline (PBS) (2:1, *w*/*v*). The mixture was stirred continuously for 2 h at room temperature and centrifuged at 6000g for 15 min at 25 °C. The supernatant was collected for further assay.

### 2.4. qPCR Primer Design

The MC housekeeping gene *β-actin* was selected as the reference gene. There is no published MC *β-actin* gene sequence in GenBank, so the partial MC *β-actin* nucleotide sequence was cloned by designing a pair of primers based on the alignment result of available Cucurbitaceae *β-actin* gene sequences (*Cucumis melo*, AB640865.1; *Cucumis sativus*, AB010922.1; and *Citrullus lanatus*, GU565958.1). The qPCR primers (MC-actin-F and MC-actin-R) for the MC-specific *β-actin* gene were designed based on the partial MC *β-actin* nucleotide sequence. The *polypeptide-P* gene specific qPCR primers were designed based on a previously published gene sequence (accession No. HQ164449) [24]. The primer design was conducted by using the Primer 3 (version 0.4.0) program (http://frodo.wi.mit.edu). 

### 2.5. qPCR Analysis

The qPCR reactions were performed on 96-well plates with the StepOnePlus™ Real-Time PCR System (Applied Biosystems, Waltham, MA, USA). The qPCR reaction mixture was prepared in a total volume of 20 μL containing: 2 μL of synthesized cDNA template, 0.8 μL of each amplification primer, 10 μl of 2 × FastStart SYBR Green Master (TaKaRa, Dalian, China), and 6.4 μL of ddH_2_O. The reaction conditions were as follows: an initial denaturation step of 95 °C for 10 min to activate the FastStart Taq DNA polymerase, followed by 45 cycles of denaturation at 95 °C for 15 s, annealing at 60 °C for 30 s, and extension at 72 °C for 30 s. Baseline and threshold cycles (Ct) were automatically determined using the StepOne™ System (Applied Biosystems, Waltham, MA, USA). Biological triplicates of each sample were used for the qPCR analysis. The relative gene expression was calculated based on the 2^−ΔΔCt^ method based on the control gene [26].

### 2.6. WB Assay

For the WB assay, a total of 100 ng protein, detected via the colloidal Coomassie Brilliant Blue (CBB) (Beyotime, Nanjing, China) method, was combined with an equal volume of 2 × SDS-PAGE loading buffer and then heated to 100 °C for 8 min and separated by 13.5% Tris-Tricine SDS-PAGE. Proteins were electroblotted onto a PVDF membrane at 100 V for 90 min using a wet blotting system (Pall Life Sciences, New York, USA). The electroblotted membranes were blocked with 5% skim milk in PBS containing 0.5% Tween-20 (PBS-T buffer) overnight at 4 °C. After washing the membranes three times with PBS-T buffer, an anti-polypeptide-P antibody [18], diluted 1:2000 in PBS-T buffer, was added and incubated for 8 h at 4 °C. Afterwards, the membranes were washed three times using PBS-T buffer. Next, the membranes were incubated with an anti-rabbit IgG secondary antibody conjugated to alkaline phosphatase (Biotium, Hayward, CA, USA) at a dilution of 1:2000 in PBS-T containing 1% skim milk for 2 h at room temperature. The signal was detected by DAB chemical chromogenic reagents (Beyotime, Shanghai, China) according to the instructions of the manufacturer.

### 2.7. Statistical Analysis

Changes in the relative expression levels of the *polypeptide-P* gene were checked for statistical significance according to Student’s *t*-test. The results were considered statistically significant if the *P*-value was <0.05. The WB results were analyzed using ImageJ2× (ij2150, Bethesda, Maryland, USA) and SPSS 19.0 software (SPSS Inc., Chicago, IL, USA).

## 3. Results and Discussion

### 3.1. qPCR Primer of MC β-actin

qPCR is a widely used method for gene transcription analysis. The housekeeping gene *β-actin* has been widely applied in qPCR analysis and is often used as an internal control. The gene alignment result of *β-actin* from three Cucurbitaceae plants (*Cucumis melo, Cucumis sativus* and *Citrullus lanatus*) displays high identity. Therefore, a pair of primers were designed to amplify this highly conserved region of the *β-actin* gene. The amplified DNA fragment was sequenced, which showed high similarity to the *β-actin* gene sequences of *Cucumis melo* (96%, AB640865.1), *Cucumis sativus* (95%, AB010922.1), and *Citrullus lanatus* (94%, GU565958.1). As a result, the primers of the MC *β-actin* were designed according to the cloned DNA sequence information and were used as an internal control for the qPCR analysis (Table 2). The specificity of designed primers was verified (data not shown).

### 3.2. Expression Analysis of the Polypeptide-P Gene in Different Accessions

In China, CB is most widely used as a health food because CB is said to contain a high content of polypeptide-P. Therefore, CB was used as a control sample in the study just to compare the relative expression levels among MCs with different accessions. The qPCR results (Figure 1A) show that the transcription expression levels of the *polypeptide-P* gene in LP, DB, RU5H, JGDR, and TCT are higher than that of the control (CB). By contrast, the transcription expression levels of the *polypeptide-P* gene in CL, TDCB, TWDC, and XC2H are lower than that of the control. RU5H displays the highest *polypeptide-P* gene expression level and XC2H has the lowest. These data indicate that the expression levels of the *polypeptide-P* gene vary among MCs with different accessions. 

Furthermore, a WB analysis was carried out to examine the expression levels of polypeptide-P in different MC accessions (Figure 1B). The results suggest that the polypeptide-P expression levels in LP, DB, RU5H, JGDR, and TCT are higher than that of the control, however, the protein expression levels in CL, TDCB, TWDC, and XC2H are much lower than that of the control, which is consistent with the qPCR results, suggesting that qPCR is a reliable and efficient method to detect the *polypeptide-P* gene expression level.

### 3.3. Expression Analysis of the Polypeptide-P Gene during Different Harvesting Seasons 

The collection of some MC tissue types was not successful, so we selected four samples (RU5H, JGDR, TDCB, and DB) to analyze the impact of harvesting seasons on expression levels of the *polypeptide-P* gene. Figure 2A indicates that the expression level in RU5H was the highest on 6 July 6 and then declined gradually in July but dropped dramatically in August and September. With regard to JGDR, TDCB, and DB, an initial expression peak of the *polypeptide-P* gene on 6 July 6 was found, but after that date, the expression levels significantly declined. These data indicate that the expression level of the *polypeptide-P* gene is time- or season-dependent and tends to be relatively high in early July, when the MC fruit is mostly immature. To testify this hypothesis, we compared *polypeptide-P* gene expression levels in the mature fruit with those in the immature fruit. Figure 2B shows that among four tested accessions of MCs, all immature fruits show higher *polypeptide-P* gene expression levels than the mature fruits, indicating that *polypeptide-P* gene transcription levels decrease with the maturity and development of fruits.

### 3.4. Expression Analysis of the Polypeptide-P Gene in Different Tissues

To examine the *polypeptide-P* gene expression change in different tissues, we systematically analyzed its expression levels in RU5H, which was harvested on 6 July, with the highest expression level of the *polypeptide-P* gene. The results in Figure 3 show that the expression pattern of the *polypeptide-P* gene in various tissues of RU5H differs greatly. High levels of transcription were observed in the leaf, fruit, and seed of RU5H. Much lower levels of *polypeptide-P* gene transcription were observed in the stem and flower than in developing seeds and fruits (Figure 3). In particular, a relatively high level of *polypeptide-P* gene transcription was found in the developing seed. Thus, the *polypeptide-P* gene shows a spatially differential expression pattern in RU5H.

## 4. Conclusions

In summary, the qPCR method established in the study provided a rapid and reliable assay system for detecting the *polypeptide-P* gene in MC. In addition, our findings show that the *polypeptide-P* gene is highly expressed in RU5H and DB compared with other accessions, which was confirmed by the result of the WB analysis. The expression pattern of the *polypeptide-P* gene was time- and tissue-dependent, and the highest expression level was found in early July, indicating an early regulation of the gene in response to the season. The *polypeptide-P* gene expression was accumulated in leaf, seed, and fruit, especially immature fruit. The present study therefore reveals that the distribution profiling of polypeptide-P and provides a reliable reference for cultivating and using high-quality hypoglycemic MCs.

## Figures and Tables

**Figure 1 genes-10-01044-f001:**
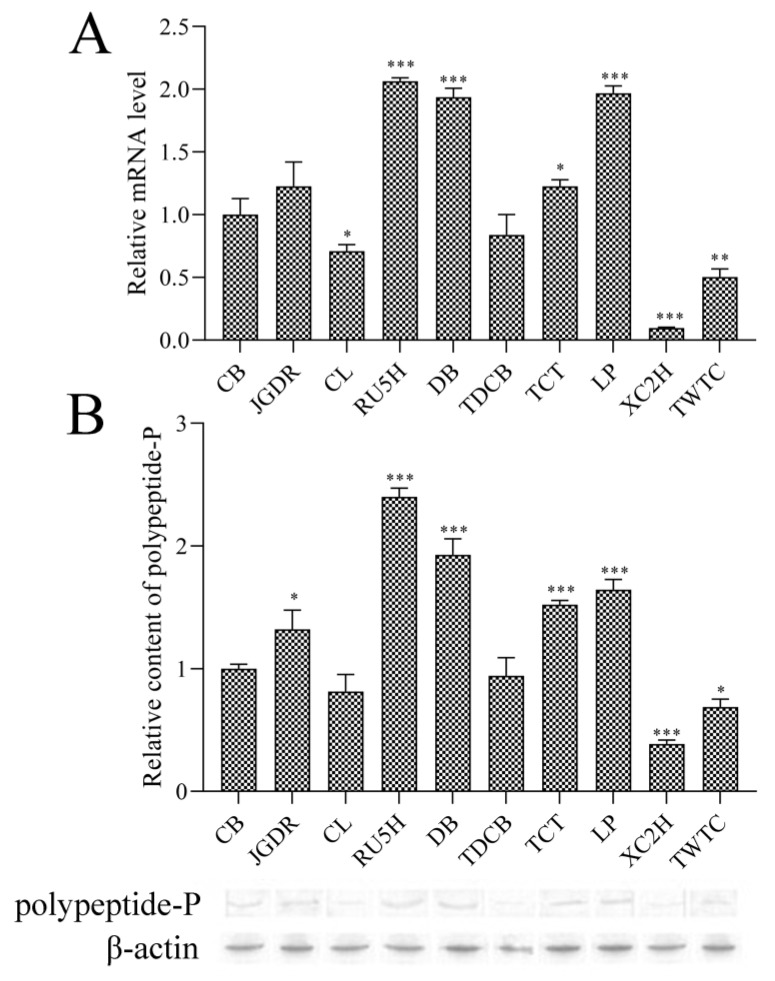
Differential expression levels of the *polypeptide-P* gene from 10 MCs with different accessions via qPCR (**A**) and Western blotting (WB) (**B**) analysis. All of the presented data are averages and standard errors (bars) of three independent experiments. *, *P* < 0.05; **, *P* < 0.01; ***, *P* < 0.001.

**Figure 2 genes-10-01044-f002:**
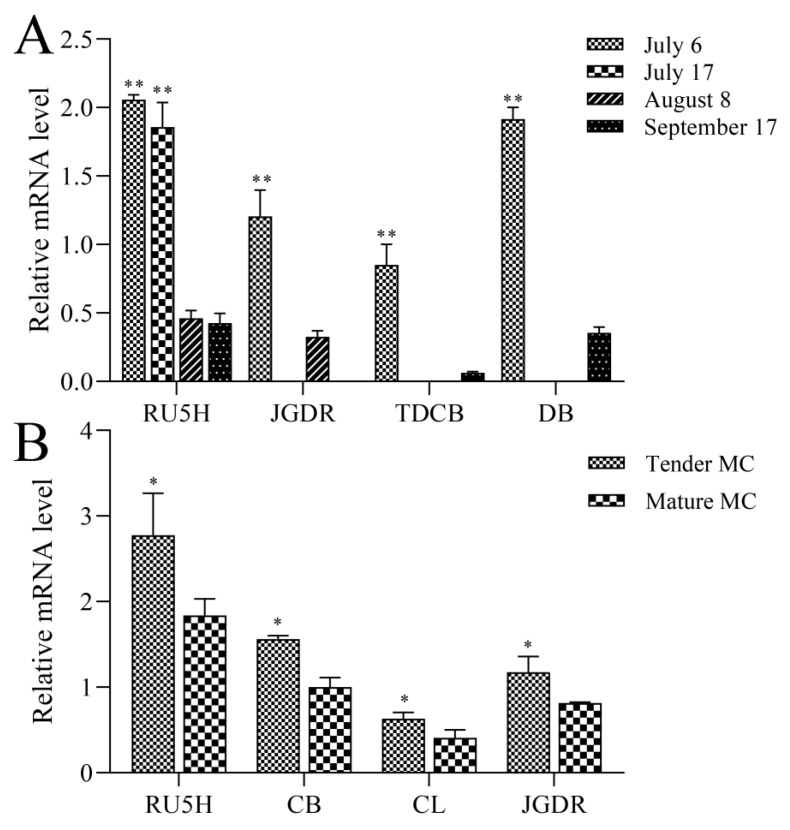
qPCR analysis of the expression levels of four different species during different harvesting seasons and maturity levels. (**A**) The expression levels of the *polypeptide-P* gene during different harvesting times: 6 July, 17 July, 8 August, and 17 September. (**B**) The expression levels of the *polypeptide-P* gene at different growth stages. All of the presented data are averages and standard errors (bars) of three independent experiments. *, *P* < 0.05; **, *P* < 0.01; ***, *P* < 0.001.

**Figure 3 genes-10-01044-f003:**
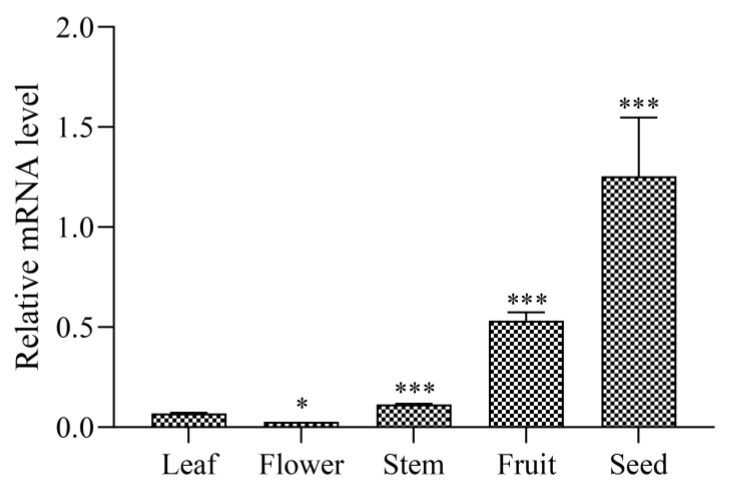
Expression levels of the *polypeptide-P* gene in different tissues of RU5H. All of the presented data are averages and standard errors (bars) of three independent experiments. *, *P* < 0.05; **, *P* < 0.01; ***, *P* < 0.001.

**Table 1 genes-10-01044-t001:** List of *Momordica charantia* (MC) accessions, origins, and fruit character.

No.	Accession	Origin ^a^	Fruit Character ^b^
1	DB	Hunan	White, medium long, continuous ridges
2	CB	Hunan	Light green, medium-long ^c^, continuous ridges
3	LP	Hunan	Green, long ^d^, discontinuous ridges
4	RU5H	Fujian	Glossy green, medium-long, discontinuous ridges
5	TDCB	Hunan	Greenish white, extra-long ^e^, discontinuous ridges
6	TWTC	Taiwan	Glossy green, extra-long, discontinuous ridges
7	CL	Guangdong	Light green, long, continuous ridges
8	XC2H	Fujian	Green, medium-long, continuous ridges
9	JGDR	Changsha	Glossy green, medium-long, discontinuous ridges
10	TCT	Guangdong	Glossy green, long, discontinuous ridges

^a^ Chinese provinces from where the accession was collected. ^b^ Fruit color, size, and surface texture. ^c^ Medium-long (10–20 cm). ^d^ Long (20–30 cm). ^e^ Extra-long (>30 cm).

**Table 2 genes-10-01044-t002:** Primers used in this study for quantitative polymerase chain reaction (qPCR) analysis.

Gene	Primer Name	Amplified Length	Primer Sequence (5′–3′)
MC *β-actin*	MC-actin-F	124bp	CCCTCCCTCATGCAATTCTC
	MC-actin-R		TCGGCAGTAGTGGTAAACATGTAAC
*polypeptide-P*	P-F	148bp	CGTGATGAAGGCAAAGTGGA
	P-R		TATCGCCAAACGGGGTAATG

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
