# Peer review of "Real-Time Quantitative PCR Analysis of the Expression Pattern of the Hypoglycemic Polypeptide-P Gene in Momordica charantia"

_genes, 2019, doi:10.3390/genes10121044_

Round 1
Reviewer 1 Report
The manuscript entitled: “Real-time quantitative PCR analysis of expression pattern of a hypoglycemic polypeptide-P in Momordica charantia during plant growth” has been revised. Authors studied the expression level of polypeptide-P in different varieties, tissues, fruit mature stages and seasons, as this fruit is traditionally used for the treatment of diabetes in southeast Asia and China. Results are very interesting and applicable.
However, I regret to inform the authors that the manuscript cannot be accepted for publication unless mayor revision was performed.
Relative quantitation assays are used to analyze changes in gene expression in a given sample relative to another reference (control) sample. Why did authors use CB as a control? There are two calculation methods for relative quantitation: Standard curve method and comparative Ct method. In the first, target quantity is determined from the standard curve and divided by the target quantity of the calibrator (in this work actine). Have authors performed the relative expression assays by this method? In this case, please, explain it in the text. If only one of the selected housekeeping genes were used, please remove information on 18S rDNA. In the comparative Ct method for relative quantitation the formula 2 -∧∧Ct is used. If that method was chosen, please indicate.
Authors published figures that, in my opinion, are not necessary but are mere controls that are performed before the real experiments. Important results begin in line 257 (Fig 6) !!!! the previous are checking’s of the good performance of the qPCR method used. Ej: lines 233-244. In some cases, the explanations of these control assays are repeated.
Some calls for tables do not correspond with their content (ej: line 154) Table 3 and table 4 could be joined. They contain repeated information. Statistical significance of the results are included in some figures (e.g. fig 7) and not in others (fig 6) It is not clear the number of samples (accessions) used in the study. In the text, authors talk about 19 samples in the abstracts and Table 1, however, 11, 8, 10 samples were used in different experiments
7. Please, consider using the same nomenclature throughout the text: Real-time PCR, real-time quantitative PCR, RT-qPCR are indistinctly used. Real-time quantitative PCR is commonly abbreviated as qPCR, and not RT qPCR (that is commonly used as retrotranscription qPCR).
English language must be deeply improved. I began to correct the English but finally, I stopped.
Discussion is not presented.
Please, find some other suggestions in the manuscript file: text, figures and tables.

Reviewer 2 Report
Although this work has been compiled and presented well, my only and biggest concern is absence of biological replications. Any conclusion drawn just based on one biological replicate doesn’t make any sense to me. How do authors justify it.
There are many inconsistencies in the amount of RNA used for cDNA preparation. For example, Legend of Fig. 6 says that RNAs were isolated from different accessions of MC and 1 mg of total RNA was used for the real-time PCR assay. Similarly it is for Fig. 7 also. 1 mg is a very high amount of RNA.
Minor comments:
Lines 54-62 and 63-71 are repeat. Please remove one paragraph.
In Methods section 2.4 “Protein isolation”: what do you mean by a simple protocol? Can you please rewrite the sentence?
Please provide GenBank accession no. of (1) partial MC-specific beta-actin sequence that was cloned in this study and (2) polypeptide P gene in methodology section 2.5 “Real-time PCR quantification PCR primer design”.
Round 2
Reviewer 1 Report
In this new version, authors have conscientiously improved the clarity of the methodology, results and discussion, although some aspects still need to be improved, including the English language
1. The methodology used in the qPCR is not yet clear to me. How much plant material do you use to make RNA extraction? How much cDNA do you use in qPCR reactions? As I mentioned in the previous revision, the concentrations of the components used in the qPCR reactions must be indicated: initial concentration and volume or alternatively, the final concentration of the components
2. Table 2 is not well understood. For example: “MC polypeptide-P” is included in the third column called “Cucurbit species”. According to the table, the first primer described is from C. melo and the second from C. sativus, and I think that is not the information that authors want to show. The first time that ssdp is mentioned is in table 2. Please, describe ssdp first in the text. I already mentioned that in the previous review (pdf) and it has not been corrected.
3. In Result section, line 141 to 145 the expression induction is indicated as percentages of induction for some accessions, and in times respect to the control for another. Please, unify for a better understanding.
